# Cost-consequence analysis of ambulatory clinic- and home-based multidrug-resistant tuberculosis management models in Eswatini

**Ernest Peresu** [1]*, **Diana De Graeve** [2], **J. Christo Heunis** [3], **N. Gladys Kigozi** [3]

**1** Centre for Development Support, Faculty of Economic and Management Sciences, University of the Free State, Bloemfontein, South Africa, **2** Faculty of Business and Economics, University of Antwerp, Antwerp, Belgium, **3** Centre for Health Systems Research & Development, University of the Free State, Bloemfontein, South Africa

\* eperesu@yahoo.com

## Abstract

### Background

We compared the cost-consequence of a home-based multidrug-resistant tuberculosis (MDR-TB) model of care, based on task-shifting of directly observed therapy (DOT) and MDR-TB injection administration to lay health workers, to a routine clinic-based strategy within an established national TB programme in Eswatini.

### Methods

Data on costs and effects of the two ambulatory models of MDR-TB care was collected using documentary data and interviews in the Lubombo and Shiselweni regions of Eswatini. Health system, patient and caregiver costs were assessed in 2014 in US$ using standard methods. Cost-consequence was calculated as the cost per patient successfully treated.

### Results

In the clinic-based and home-based models of care, respectively, a total of 96 and 106 MDR-TB patients were enrolled in 2014, with treatment success rates of 67.8% and 82.1%. Health system costs per patient treated were slightly lower in the home-based strategy (US $19 598) compared to the clinic-based model (US$20 007). The largest costs in both models were for inpatient care, administration of DOT and injectable treatment, and drugs. Costs incurred by patients and caregivers were considerably higher in the clinic-based model of care due to the higher direct travel costs to the nearest clinic to receive DOT and injections daily. In total, MDR patients in the clinic-based strategy incurred average costs of US$670 compared to US$275 for MDR-TB patients in the home-based model. MDR-TB patients in the home-based programme, where DOT and injections was provided in their homes, only incurred out-of-pocket travel expenses for monthly outpatient treatment monitoring visits averaging US$100. The cost per successfully treated patient was US$31 106 and US$24 157 in the clinic-based and home-based models of care, respectively. The analysis showed

**Data Availability Statement:** The data underlying the results presented in the study are included in

this published article and has been shared in supporting files S1 and S2.

**Funding:** The authors received no specific funding for this work.

**Competing interests:** The authors have declared that no competing interests exist.

**Abbreviations:** CP, continuation phase; CTS, community treatment supporter; DOT, directly observed therapy; HIV, human immunodeficiency virus; IP, intensive phase; IUATLD, International Union Against Tuberculosis and Lung Disease; MOH, Ministry of Health; MSF, Médecins Sans Frontières; MDR-TB, multidrug-resistant tuberculosis; NTCP, National Tuberculosis Control Programme; PHC, primary health care; SDG, Sustainable Development Goal; TB, tuberculosis; WHO, World Health Organization.

that, in addition to the health benefits, direct and indirect costs for patients and their caregivers were lower in the home-based care model.

## Conclusion

The home-based strategy used less resources and generated substantial health and economic benefits, particularly for patients and their caregivers, and decision makers can consider this approach as an alternative to expand and optimise MDR-TB control in resource-limited settings. Further research to understand the appropriate mix of treatment support components that are most important for optimal clinical and public health outcomes in the ambulatory home-based model of MDR-TB care is necessary.

## Background

The global strategy to control tuberculosis (TB) has been highlighted as a priority within the framework of the End TB Strategy and the Sustainable Development Goals (SDGs) agendas [1, 2]. Nevertheless, multidrug-resistant TB (MDR-TB) treatment is accessible to only one-third of the estimated cases globally [3]. In Eswatini, access to MDR-TB care has remained limited, reflecting a range of interconnected issues including the widespread shortage of professional frontline healthcare workers, poor geographic accessibility of health facilities and high costs incurred by patients [4–6]. Unsurprisingly, in 2019, the country reported a suboptimal MDR-TB treatment success rate of 72% (2016 cohort) which was below the World Health Organization (WHO) target of 75% or higher [3, 7].

To reduce costs to the health system, expand treatment access and provide patient-centred care, the Eswatini National TB Control Programme (NTCP) adopted the WHO ambulatory clinic-based care model for managing MDR-TB in three of the country's four regions [8]. In the remaining region, MDR-TB care was delivered using a novel patient-centred home-based treatment approach based on task-shifting of directly observed therapy (DOT) and intramuscular MDR-TB injection administration–traditionally restricted to professional nurses–to lay health workers, referred to as community treatment supporters (CTSs) [4]. Although the WHO has prioritised the use of shorter and fully oral MDR-TB treatment regimens (nine to 12 months instead of the usual two years), at least for the time being, the use of long injectable-containing regimens in settings with high rates of complex drug resistance like Eswatini persists [5, 9]. Instead of making trips to the nearest clinic, this model of care allowed MDR-TB patients to receive DOT and daily injections during the first eight months of treatment from CTSs in their (the patients') homes [6, 10].

Previous economic evaluation studies have consistently demonstrated the cost-effectiveness of ambulatory models of care embedded within communities, closer to or in the patient's home, rather than hospitalised care [11–13]. Yet, to date no economic evaluation of the ambulatory clinic- and home-based care models has been conducted in Eswatini. We describe an assessment of the costs and impact on treatment success of routine ambulatory clinic-based MDR-TB care in comparison to home-based care established in two regions of Eswatini. Cost-consequence analysis allows decision makers to consider the costs and resource consequences resulting from, or associated with, the two models of care, as well as considering relevant clinical benefits alongside the cost analysis. This current study adds to the limited literature and provides results from an Eswatini context.

## Methods

### Study design and setting

We evaluated, firstly, the clinic-based approach implemented by the Ministry of Health (MOH) in the Lubombo region and, secondly, the home-based model adopted by Médecins Sans Frontières (MSF), in the Shiselweni region within the NTCP. The evaluation considered a cohort of patients enrolled into MDR-TB care between 1 January and 31 December 2014 over a two-year analytic horizon.

Lubombo and Shiselweni form part of the country's four administrative regions. The two regions have relatively similar socio-economic conditions [14]. The Shiselweni region has one referral hospital and two health centres that support 18 smaller primary health care (PHC) clinics that form part of the regional health network managed by the MOH. MDR-TB patients requiring inpatient care are hospitalised in the TB ward at one of the health centres. The Lubombo region has an estimated population of about 212 531 and a health facility to population ratio of about 1 per 4 500 [14]. Health facilities involved in MDR-TB care in the Lubombo region comprise of a mission hospital and a rural health centre that relate closely to the National TB Hospital in Manzini, the country's referral hospital for all forms of TB. DOT and out-patient injection administration is accessible from all the PHC facilities in the region. The prevalence of MDR-TB across the two regions illustrates negligible dissimilarity in distribution [5]. HIV prevalence among the general population is estimated at 27.0%, with no significant geographic variation across the regions of the country [5].

### Alternative strategies

In Eswatini, patients receive the full standardised MDR-TB regimen on an ambulatory basis at their nearest outpatient health care facility. Ambulatory care can either be clinic-based (patient travels to the clinic five days per week to receive DOT and injections from a healthcare worker) or home-based (patient receives DOT and injections seven days per week in their homes from a CTS). The main components and activities of each of the alternative strategies are summarised in Table 1; detailed descriptions are available elsewhere [6, 10]. The MDR-TB treatment regimen consisted of an eight-month intensive phase (IP) based on one injectable and four oral drugs followed by a 16-month continuation phase (CP) with the same drugs, less the injectable. Besides the differences in some treatment support components, patients in both regions received the same clinical evaluation, diagnosis and treatment according to the national MDR-TB treatment guidelines.

### Cost and cost-consequence analysis

The economic analysis was based, broadly, on a retrospective analysis of patient-level data, and encompassed elements of health system, patient and caregiver perspectives using current standards for cost-consequence analysis [15]. Health system costs, and patient and caregiver costs were estimated using data obtained from the 2014 and 2017 heterogenous cohorts in the two regions, respectively. All costs were expressed in year 2014 United States dollar ($) currency.

### Health system costs

For health systems costs, the average cost of each component of treatment was calculated by combining data on the quantity of resources used with the unit prices, with the exception of non-salary recurrent expenditure that was only available in aggregated form. Health system costs consisted of costs of inpatient stay, outpatient costs associated with laboratory and radiology tests; outpatient attendance; drugs and medical supplies (including for adverse effects),

**Table 1. Comparison of alternative strategies.**

| Treatment component | Clinic-based strategy | Home-based strategy |
|---|---|---|
| | Lubombo region, MOH | Shiselweni region, MSF |
| Hospital admission | For treatment initiation and often prolonged if patient's home is distant from nearest clinic, unstable clinical condition, unfavourable social and logistic conditions **Remove this empty row** | Only if necessitated by clinical condition |
| Caregiver/treatment supporter | Family member | IP–CTS (neighbour) CP–family member |
| Training of treatment supporter | Routine training on DOT and household IPC | Training provided specifically to CTSs for injection administration, DOT and household IPC. Family member trained on DOT and household IPC |
| DOT supervisor | Nurse supervised morning DOT during IP Family caregiver supervised evening DOT during IP and CP | CTS in IP and option of family member during CP |
| Injection administration | Travels 5 days per week to nearest clinic for injection administration by a nurse during IP | Receives intramuscular injections 7 days per week from a CTS in their (patient's) home during IP. |
| Out-patient visits to health facility for review and collection of drugs | Monthly in company of a family member | IP–monthly in company of a CTS CP–monthly in company of a CTS/family member |
| Community supervision | Community MDR-TB nurses once a month during IP | Community MDR-TB nurses twice a month during IP and once a month during CP |
| Financial incentives | N/A | Patients received US$41/month during IP and US$8/month during CP CTS received US$58/month during IP |
| Food enablers | N/A | Patients received food packages valued at US$33 every month throughout treatment |
| Structural environmental household TB infection prevention and control measures | N/A | Environmental control interventions aimed at improving natural ventilation such as structural addition of windows in patient's room or in some instances, construction of a new one-roomed house for patient to sleep alone |

MOH–Ministry of Health; MSF–Médecins Sans Frontières; DOT–directly observed therapy; N/A–not applicable; CP–continuation phase; CTS–community treatment supporter; IP–intensive phase; IPC–infection prevention and control; US$–United States dollars

DOT and injection administration at the clinic or in the home; training and supervision visits of CTSs and overall programme management (including costs of structural adaptations to the patient's home).

For all strategies, data sources included primary usage data abstracted from patient medical records, central medical stores, expenditure and programme reports, vehicle logbooks, and face-to-face interviews with staff from the MOH and MSF. Staff costs were estimated from standard salary and benefits scales for established positions obtained from the MOH. Capital costs for items such as buildings, vehicle and equipment were depreciated using standard recommended methods and a discount rate of 3% [15]. Joint costs were allocated based on the proportion of time the cost item was used for MDR-TB programme components and activities.

Recurrent overhead costs for elements such as fuel, maintenance of buildings and vehicles were obtained from expenditure and programme reports, vehicle logbooks, and face-to-face interviews with staff from the MOH and MSF [S2 Appendix: MDR-TB electronic medical records abstraction tool]. Costs for drugs were estimated based on the Central Medical Stores price list at the time. Visits for injections during the intensive phase were allocated for five and seven days a week for the clinic-based and community-based model of care respectively. The total programme cost per treatment component was calculated as the unit cost of each treatment component multiplied by the number of times this cost was incurred in each region. A supporting information file [S1 Table: Unit costs of components associated with diagnosis and

treatment of MDR-TB] shows a unit cost table outlining the published unit costs used in this cost-consequence analysis and their sources–mainly from a South African context [16].

## Patient and caregiver costs

Patient and caregiver costs were estimated using a survey among a purposive sample of 78 patients under each of the care models in May 2017. A validated structured questionnaire was used to gather retrospective information about study participants' use of health care services [S1 Appendix: MDR-TB-related costs incurred during treatment]. Travel time, mode of transport and patient/caregiver time off from work for the health facility follow-up visits were collected using the questionnaire, and responses were taken to be representative of the patient and caregiver for all subsequent visits [17].

Indirect costs were defined as earnings lost by MDR-TB patients and their caregiver while seeking and undergoing and supervising or administering treatment. We estimated opportunity costs for all ambulatory medical visits by ascertaining the patient's time lost due to MDR-TB illness, including travelling, outpatient visits, admission in hospital, and absence from work [S3 Appendix: File Medical Records Abstraction Data]. CTSs and family attendants' time devoted to caregiving was estimated from the outpatient visits a carer accompanied the patient to the health facility and visited a patient during admission in hospital. The survey also assessed other costs incurred and time spent by CTSs attending training, supervising DOT and administering intramuscular MDR-TB injections. To calculate the opportunity cost of time, the number of hours lost was multiplied by the estimated hourly wage derived from the country's weekly minimum wage [18]. Total MDR-TB treatment costs for patients and caregivers were estimated by extrapolating the monthly costs according to standard recommended durations of the intensive and continuation phases: eight months and 16 months respectively. This information was combined with other sources of reference unit costs in order to calculate a mean cost of service costs per participant per model of care for the cost-consequences analysis.

Given that the target population group, MDR-TB patients, in both models of care was small, we choose to study the entire adult population of MDR-TB patients in the two regions. Patients were eligible to participate in the study if they were aged 18 years or older, had completed at least one month of MDR-TB treatment and were being treated at a health facility in the selected regions. The study excluded MDR-TB patients who had profound deafness from medication toxicity which resulted in difficulties in comprehension and completion of the interviews.

Trained interviewers conducted face-to-face interviews in either siSwati or English with participants attending the MDR-TB treating facilities for their monthly review in the two regions. All patients and their accompanying caregiver were approached at the end of their follow-up visit and referred to a trained interviewer stationed in a private room within the MDR-TB unit at the health centre. Participation in the study was voluntary and not linked to patients' care or CTS' job security. No incentives were offered to induce participation. Written informed consent was obtained from patients and their caregivers prior to participation in the interviews. To reduce recall bias, only costs related to the previous month were collected.

## Outcome measures

The primary outcomes measure for the economic evaluation were: 1) cured, 2) completed treatment, 3) died, 4) defaulted, 5) transferred out of the region, and 6) failed treatment. Successful treatment rates were obtained from the standard recording and reporting system used by the NTCP which followed the International Union Against Tuberculosis and Lung Disease

(IUATLD) guidelines [19]. For each strategy, the average cost per patient successfully treated was calculated.

### Ethical clearance and authorisation

Ethical approval was obtained from the Scientific and Ethics Committee of Eswatini and the Health Sciences Research Ethics Committee (IRB00006240), University of Free State. Authorisation of the study was granted by the NTCP and MSF.

## Results

### Patient characteristics

In this study, a total of 202 MDR-TB patients' records were reviewed, 96 under the clinic-based strategy in the Lubombo region and 106 in the home-based care model in the Shiselweni region (Table 2). The mean age of MDR-TB patients in the clinic-based (34 years) and home-based (35 years) models was almost similar. Overall, a significant proportion of MDR-TB patients in the clinic-based approach (n = 74; 77.1%) and home-based strategy (n = 87; 82.1%) were unemployed. More than three-quarters of patients in each of the models of care had MDR-TB/human immunodeficiency virus (HIV) co-infection. A chi-square test showed no statistically significant difference between the socio-demographic characteristics of patients in the two regions; gender (p = 0.534), mean age (p = 0.584), employment status (p = 0.378), co-morbid chronic medical condition (p = 0.232) and HIV co-infection (p = 0.585).

### Overall treatment details

The frequency of outpatient treatment monitoring visits, and DOT and injection administration at the clinic or in the home and supervision visits varied according to total duration of treatment, length of the injectable phase and number of inpatient hospitalisation days across the two models (Table 3).

### Health system costs

From a societal point of view, health system costs per patient treated were slightly lower in the home-based strategy (US$19 598) compared to clinic-based model (US$20 007) (Table 4). The largest costs in both models were for inpatient care, administration of DOT and injectable treatment, and drugs (approximately one-quarter of the total health system costs in each model). The total cost to provide the laboratory diagnostic and monitoring tests was US$480 per patient for the clinic-based model and US$1 881 per patient for the home-based strategy.

**Table 2. Socio-demographic and clinical characteristics of MDR-TB patients.**

| Characteristics | Clinic-based strategy | Home-based strategy | P value |
|---|---|---|---|
|  | N = 96 (%) | N = 106 (%) |  |
| Female | 42 (43.7) | 51 (48.1) | 0.534 |
| Mean age (years) | 34 | 35 | 0.584 |
| Unemployed | 74 (77.1) | 87 (82.1) | 0.378 |
| Chronic medical condition/s | 21 (21.9) | 31 (29.2) | 0.232 |
| HIV-infected[a] | 73 (76.0) | 84 (79.2) | 0.585 |

[a]MDR-TB and HIV services were integrated, and all MDR-TB/HIV co-infected patients were enrolled for antiretroviral treatment

**Table 3. Summary of key treatment management indicators for MDR-TB patients.**

| Treatment component | Lubombo | | Shiselweni | |
| --- | --- | --- | --- | --- |
| | Clinic-based | | Home-based | |
| | Strategy (N = 96) | | Strategy (N = 106) | |
| | Mean | 95% CI | Mean | 95% CI |
| Duration of MDR-TB treatment (days) | 600 | 562–538 | 641 | 631–686 |
| Duration of intensive phase (days) | 235 | 228–242 | 240 | 235–244 |
| Duration of continuation phase (days) | 364 | 331–398 | 419 | 395–443 |
| Inpatient care (days) | 41 | 35–47 | 23 | 20–27 |
| Injections administered at clinic | 142 | 139–144 | N/A | N/A |
| Injections administered at home | N/A | N/A | 216 | 211–222 |
| Supervision visits by community MDR-TB nurse | 23 | 18–28 | 32 | 25–39 |
| Hospital OPD visits for treatment monitoring/collection of drugs | 23 | 19–27 | 24 | 18–30 |
| DOT supervision by family member | 369 | 335–404 | 418 | 393–442 |

OPD–outpatient department; DOT–directly observed therapy; confidence interval

The frequency of laboratory tests varied in the two models and was lower in the clinic-based model than in the home-based strategy.

## Patient and caregiver costs

Costs incurred by patients c were considerably higher–more than double–in the clinic-based model of care due to the higher direct travel costs to the nearest clinic to receive DOT and injections daily. Travel costs accounted for about 43 percent and 63 percent of the total costs incurred by patients and caregivers in the clinic-based model, respectively (Tables 5 and 6). In total, MDR patients in the clinic-based strategy incurred average costs of US$670 compared to US$275 for MDR-TB patients in the home-based model. MDR-TB patients in the home-based programme, where DOT and injections was provided in their homes, only incurred out-of-pocket travel expenses for monthly outpatient treatment monitoring visits averaging US$100. Opportunity costs of travel and treatment time for patients ranged from US$239 in the clinic-based model to US$120 in the home-based approach.

**Table 4. Health system costs of managing MDR-TB patients from diagnosis to completion of treatment.**

| Cost component | Lubombo | Shiselweni | Difference (US$) |
| --- | --- | --- | --- |
| | Clinic-based strategy | Home-based strategy | |
| | Cost US$ (%) | Cost US$ (%) | |
| Inpatient care | 6 414 (32) | 3 663 (19) | |
| Laboratory and radiology tests | 480 (2) | 1 881 (10) | |
| Drugs | 4 509 (23) | 5 512 (28) | |
| Hospital OPD treatment monitoring | 2 056 (10) | 2 007 (10) | |
| Clinic DOT/injection administration visit | 3 399 (17) | 0 (0) | |
| DOT and injection administration by CTS | 0 (0) | 2 379 (12) | |
| DOT supervision by family member/CTS | 1 478 (7) | 1 670 (9) | |
| Supervision by community MDR-TB nurse | 446 (2) | 621 (3) | |
| Programme level costs | 1 225 (6) | 1 865 (10) | |
| *Total health system costs* | *20 007 (100)* | *19 598 (100)* | *-409* |

Notes: All costs were adjusted to 2014 prices. OPD–outpatient department; DOT–directly observed therapy; CTS–community treatment supporter

**Table 5. Opportunity costs and direct costs for patients.**

| | Clinic-based strategy | Home-based strategy |
|---|---|---|
| | Cost US$ (%) | Cost US$ (%) |
| **Opportunity costs** | | |
| Hospitalisation | 101 (15) | 58 (21) |
| Treatment follow-up visits | 28 (4) | 28 (10) |
| Clinic visits for DOT/injections | 110 (16) | 0 (0) |
| Receiving DOT/injections in the home | 0 (0) | 34 (12) |
| **Direct costs** | | |
| Transport expenses to hospital OPD visits | 95 (14) | 100 (36) |
| Cost of food during visit to hospital OPD visits | 51 (8) | 56 (20) |
| Transport expenses for DOT/injection administration visit | 285 (43) | 0 (0) |
| *Total costs for patients* | *670 (100)* | *276 (100)* |

## Treatment outcomes

Outcomes according to treatment strategy are displayed in Table 7. Among the 96 patients initiating treatment in the clinic-based model, 35 (36.5%) were cured, 30 (31.3%) completed treatment, seven (7.3%) failed, 13 (13.5%) died and 11 (11.5%) defaulted. Of the 106 patients treated in the community-based model, 69 (65.1%) were cured and 18 (17.0%) completed treatment. The overall treatment success rate was higher in the home-based approach (82.1%) compared to the clinic-based model (67.8%), successfully treating nearly 15% more of the cases (Table 7).

## Costs and treatment outcomes

The home-based model of care had superior patient outcomes (82.1%) compared to the clinic-based strategy (67.1%), incurred lower costs per patient treated and per patient successfully treated of US$1 080 and US$6 790 respectively (Table 8). The total cost per successfully treated patient in the home-based strategy was US$24 488, approximately 22% lower than in the clinic-based model.

## Sensitivity analysis

Sensitivity analyses were performed to explore the degree of uncertainty of health system and patient costs to variations in length of hospitalisation, duration of treatment and the frequency

**Table 6. Opportunity costs and direct costs for caregivers.**

| | Clinic-based strategy | Home-based strategy |
|---|---|---|
| | Cost US$ (%) | Cost US$ (%) |
| **Opportunity costs** | | |
| Training attendance | 1 (0) | 11 (6) |
| Accompanying patient for follow-up visits | 7 (2) | 7 (4) |
| Accompanying patient to clinic for DOT/injections | 44 (10) | 0 (0) |
| Administering DOT/injections | 0 (0) | 34 (20) |
| DOT supervision | 19 (4) | 22 (13) |
| **Direct costs** | | |
| Transport expenses accompanying patient to follow-up visits | 95 (21) | 100 (57) |
| Transport expenses accompanying patients to clinic for DOT/injection administration visits | 285 (63) | 0 (0) |
| *Total costs for caregivers* | *451 (100)* | *174 (100)* |

**Table 7. Treatment outcomes of MDR-TB patients in alternative strategies.**

| Treatment outcome | Lubombo | Shiselweni |
|---|---|---|
| | Clinic-based strategy | Home-based strategy |
| | n (%) | n (%) |
| Total number of patients in cohort | 96 (100.0) | 106 (100.0) |
| Number of patients cured | 35 (36.5) | 69 (65.1) |
| Number of patients completed treatment | 30 (31.3) | 18 (17.0) |
| Number of patients failed | 7 (7.3) | 6 (5.7) |
| Number of patients died | 13 (13.5) | 9 (8.5) |
| Number of patients defaulted | 11 (11.5) | 4 (3.8) |
| Total patients successfully treated[a] | 65 (67.7) | 87 (82.1) |

[a]Treatment success is the proportion of patients in whom the treatment outcome was either cured or completed.

of laboratory tests. We performed a one-way sensitivity analysis deflating the length of inpatient care and number of visits for DOT and injection administration (Table 9). We also approximated an upper bound for laboratory test utilisation by assuming both strategies followed national guidelines.

In one-way sensitivity analysis, health system costs varied with the length of stay in hospital and were moderately sensitive to the assumption regarding increases in laboratory testing and total number of DOT and injection administration visits (Table 9). Patient costs were mildly sensitive to length of inpatient care and number of visits to health facility or caregiver for DOT supervision and injection administration.

## Discussion

As far as could be ascertained, this is the first study to evaluate the cost-consequence of an ambulatory approach using CTSs to administer DOT and MDR-TB injectable medicines in patients' homes. Our analysis indicates that the home-based strategy was less costly, and had considerably superior treatment outcomes than the clinic-based MDR-TB model of care in a high HIV prevalence setting. In this study, the cost per successfully treated MDR-TB patient in the home-based model was lower (US$24 488) compared to clinic-based care (US$31 278). However, the programmatic implementation of the novel home-based model of care that relies on the task-shifting of highly differentiated clinical tasks such as injection administration to CTSs is not without challenges. Despite acceptability by MDR-TB patients, concerns about suboptimal quality of clinical care, patient safety and malpractice liability fears related to the use of lay health workers have been expressed previously [10, 20].

Future research could explore how the different MDR-TB care strategies can be complementary and implemented together in the same setting. From a programmatic perspective,

**Table 8. Summary of costs and consequences.**

| | Clinic-based strategy | Home-based strategy | Difference |
|---|---|---|---|
| Health system costs | 20 007 | 19 598 | |
| Patient costs | 670 | 276 | |
| Caregiver costs | 451 | 174 | |
| Cost per patient (US$) | 21 128 | 20 048 | -1 080 |
| Success rate (%) | 67.1 | 82.1 | 15 |
| Average cost per success (US$) | 31 278 | 24 488 | -6 790 |

**Table 9. Sensitivity analyses for varying differences of length of hospitalisation, laboratory tests and total number of DOT/injection visits.**

| | Health system costs | | Patient costs | |
|---|---|---|---|---|
| | Clinic-based strategy (US$) | Home-based strategy (US$) | Clinic-based strategy (US$) | Home-based strategy (US$) |
| Length of hospitalisation (days) | | | | |
| 20 days | 16 733 | 19 074 | 618 | 266 |
| *Base case* | *20 007* | *19 598* | *670* | *276* |
| Laboratory tests | | | | |
| *Base case* | *20 007* | *19 598* | | |
| 100% | 22 533 | 20 045 | | |
| Total number of DOT/injection visits | | | | |
| 112 visits (3 weekly visits x 8 months + 1 monthly visits x 16 months) | | | 647 | 251 |
| *176 visits (5 weekly visits x 8 months + 1 monthly visits x 16 months)–base case* | | | *670* | *276* |

this would enable patients for whom one of the strategies is otherwise inappropriate to choose and move seamlessly between the different MDR-TB models of care based on their own needs and differing community contexts.

In both models, drug costs were a major contributor to the total health system costs –23% and 28% in the clinic-based and home-based model respectively. Drawing from WHO recommendations, the NTCP has stated its intention to expand the roll out of shorter-duration regimens contingent on patient preference, clinical judgement and results of drug susceptibility testing that will have substantial cost implications [9, 21]. Shortened treatment regimens imply reduced periods of care requiring fewer drugs and laboratory tests, less visits to health facilities and, ultimately, reduced economic burden on households.

Patients in the clinic-based model had generally longer periods of hospitalisation (41 days), particularly during the intensive phase, than those under home-based care (23 days). Consequently, inpatient care was the main contributor to health system costs (32%) in the clinic-based model. Our analysis did not consider the severity of the disease at the time of treatment initiation or delays in initiating MDR-TB treatment after diagnosis. This relatively higher rate of hospitalisation may be taken as a proxy measure of severe disease presentation by patients in the Lubombo region. The elevated risk of unfavourable outcomes such as high mortality and increased person-to-person transmissions among those patients who delayed initiating MDR-TB treatment has been consistently reported in studies from elsewhere [22].

In the clinic-based strategy, MDR-TB patients and their caregivers incurred substantial direct out-of-pocket travel costs (US$285 [43%]) in addition to the associated indirect costs (US$239) from lost time accessing daily injections at their nearest health facility. These costs are recognised barriers to treatment completion [23, 24]. While the fact that patients incur significant opportunity costs when seeking care in the clinic-based model may not be surprising, quantifying these opportunity costs illuminates a hidden piece of health care spending by patients and caregivers. Much of these opportunity costs were due to time spent in activities (travelling to the nearest health facility) other than actually receiving care. These opportunity costs may be more readily considered by policy makers in evaluating alternative strategies to improve the efficient delivery of patient-centred care and eliminating non-direct patient care time.

Patient direct out-of-pocket costs–from diagnosis to treatment and ultimately cure–are also, in part, a function of the structure of the public health system, with transport costs

reflecting accessibility of MDR-TB services and distribution of health facilities. In the home-based model, patients collected their monthly supply of oral and injectable drugs during the outpatient treatment monitoring visit and CTSs administered the daily treatment in their (patients') homes, requiring no travel to the clinics for injections and limited supervisory visits by professional healthcare workers. Implicitly, the home-based approach may free up resources and allow professional healthcare workers to focus on other important healthcare tasks in TB control.

The serious downstream consequences of the catastrophic cost of illness for MDR-TB patients are well documented and comprise of non-adherence to care, treatment failure and increased risk of onward transmission of the disease in the community [23, 24]. A disproportionately high proportion of patients in this study, 77% in the clinic-based model and 82% in the home-based approach, had no source of income. The End TB Strategy recognises social protection interventions as powerful tools for mitigating the catastrophic costs experienced by MDR-TB-affected households and optimising MDR-TB control indicators [7]. In the home-based strategy, patients received monthly incentives of US$74 and US$41 during the IP and CP, respectively. CTSs also received a monthly stipend of US$58 during the IP only. Although these cash transfer payments could not be included in the analysis from a societal point of view, they were still a cost burden for the health care system (US$1 718) and were substantial relative to the total costs incurred by patients and their caregivers.

Supplementing standard MDR-TB care with a mix of incentives may have been decisive in mitigating the catastrophic costs experienced by MDR-TB-affected households and optimised treatment success observed in the home-based strategy [6, 25]. However, it remains unclear to what extent, if any, the superior treatment success observed in the home-based strategy can be attributed to the availability of the social protection packages. Further research can evaluate the influence of these social protection packages on patient healthcare expenditures and treatment outcomes.

In this study, treatment success rate was higher in the home-based approach (82%) than clinic-based model (68%), comparing very well with the MDR-TB treatment success rate of 56% recorded in 2018 globally [3]. These findings are consistent with recent evidence on the feasibility of ambulatory home-based care models [13, 26]. Interestingly, treatment success rates from our study were superior or comparable to published outcomes from a similar setting in South Africa with a high prevalence of HIV [13]. In the South African context, some of the community-based care models used nurse-led mobile injection teams for daily DOT supervision and injection provision in patients' homes. These unexpected differences can possibly be attributed to the high HIV treatment uptake (n = 157; 100%) in the present study. Among MDR-TB patients co-infected with HIV, antiretroviral therapy is an important contributor of treatment success [26, 27].

The limitations of this economic analysis include the retrospective approach, which relied upon interviews and medical records of two different cohorts to determine the resource categories and volume of resource use in each model of care. Patients may have failed to accurately recall certain expenditures, for example, the precise number of times a health service was utilised and the time spent seeking care. However, we consider that any effect on costs is likely to be modest and may have affected each alternative in a similar fashion.

The design of this study–small purposive sample and non-randomisation–restricted the economic analysis to a cost-consequence analysis, which is necessarily narrower in scope than alternative economic evaluation study designs. Nevertheless, our analysis presents disaggregated costs, and treatment outcomes that allow decision makers to form their own opinion on the two models of care under comparison. Another possible limitation was that we had no data on unit costs for some health system resources such as laboratory investigations in

Eswatini where the two ambulatory models of care were implemented. However, the negligible variation in MDR-TB clinical practices, life expectancy, payment systems and discount rate enabled the geographic transferability of cost data reported by Pooran and colleagues in South Africa to the Eswatini context [16].

## Conclusion

Our cost-consequence analysis fills a unique gap in the literature as the first study to assess the cost implications of two different models of MDR-TB care in a resource-limited setting and provides initial evidence to inform future research. Overall, our study showed that the ambulatory home-based strategy uses less resources, generated substantial health and economic benefits, particularly for patients and caregivers, and decision makers can consider this approach as an alternative to the clinic-based model of care.

However, to support the scaling up of universal access to uninterrupted MDR-TB care in Eswatini and the achievement of national targets, the SDGs and post-2015 global heath targets, it is imperative for policy makers and programme managers to consider reorienting MDR-TB management by strengthening care in the community and replicating the conditions that may have contributed to its success in the home-based strategy. More work remains to be done to better understand the appropriate mix of treatment support components that are most important for optimal clinical and public health outcomes in the ambulatory home-based model of MDR-TB care.

## Supporting information

**S1 Table. Unit costs of components associated with diagnosis and treatment of MDR-TB.**
(DOCX)

**S1 Appendix. MDR-TB-related costs incurred during treatment.**
(DOCX)

**S2 Appendix. MDR-TB electronic medical records abstraction tool.**
(DOCX)

**S3 Appendix. File medical records abstraction data.**
(XLSX)

## Acknowledgments

The authors thank the National Tuberculosis Control Programme and Médecins Sans Frontières for authorising the research. We sincerely thank all stakeholders, community MDR-TB nurses and treatment supporters in the Shiselweni region for their support and participation in this study.

## Author Contributions

**Conceptualization:** Ernest Peresu, Diana De Graeve, J. Christo Heunis, N. Gladys Kigozi.

**Data curation:** Ernest Peresu.

**Formal analysis:** Ernest Peresu.

**Investigation:** Ernest Peresu.

**Resources:** Ernest Peresu.

**Supervision:** Diana De Graeve, J. Christo Heunis, N. Gladys Kigozi.

**Validation:** Diana De Graeve, J. Christo Heunis, N. Gladys Kigozi.

**Visualization:** Ernest Peresu.

**Writing – original draft:** Ernest Peresu.

**Writing – review & editing:** Ernest Peresu, Diana De Graeve, J. Christo Heunis, N. Gladys Kigozi.

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
