## [Decision Letter · Decision Letter 0]

31 Aug 2022

PONE-D-22-05265Cost-effectiveness of ambulatory clinic- and home-based multidrug-resistant tuberculosis management models in EswatiniPLOS ONE

Dear Dr. Peresu,

Thank you for submitting your manuscript to PLOS ONE. Firstly, we would like to apologize for the delay in processing your manuscript. It has been exceptionally difficult to secure reviewers to evaluate your study. We have now received one completed review, which is available below. The reviewer has raised significant scientific concerns about the study that need to be addressed in a revision. Please note that we have only been able to secure a single reviewer to assess your manuscript. We are issuing a decision on your manuscript at this point to prevent further delays in the evaluation of your manuscript. Please be aware that the editor who handles your revised manuscript might find it necessary to invite additional reviewers to assess this work once the revised manuscript is submitted. However, we will aim to proceed on the basis of this single review if possible.  After careful consideration, we feel that it has merit but does not fully meet PLOS ONE’s publication criteria as it currently stands. Therefore, we invite you to submit a revised version of the manuscript that addresses the points raised during the review process.

We look forward to receiving your revised manuscript.

Kind regards,

Miquel Vall-llosera Camps

Senior Editor

PLOS ONE

Journal Requirements:

Reviewers' comments:

Reviewer's Responses to Questions

**Comments to the Author**

1. Is the manuscript technically sound, and do the data support the conclusions?

Reviewer #1: Partly

2. Has the statistical analysis been performed appropriately and rigorously? 

Reviewer #1: No

3. Have the authors made all data underlying the findings in their manuscript fully available?

Reviewer #1: No

4. Is the manuscript presented in an intelligible fashion and written in standard English?

Reviewer #1: Yes

5. Review Comments to the Author

Reviewer #1: PONE-D-22-05265

In this study the authors assessed and compared costs, from the societal perspective, of a home-based and routine MDR-TB care models in Lubombo and Shiselweni regions of Eswatini, a country with high MDR-TB burden. The authors report that the home-based care enabled by task-shifting of DOTS and injection therapy part of MDR-TB treatment by lay health workers reduced costs (both health service and patient perspective) and increased MDR-TB treatment success rates compared to the status quo.

Overall assessment

The research topic explored by the authors of this manuscript is an important area where there is not a lot of empiric evidence. The authors provide a good range of empiric evidence on costs using a wide range of data sources where costs can be ascertained at individual patient level from the societal perspective (partial). However, this reviewer has several major concerns for 1) the methodology in evaluating the ‘cost-effectiveness’, 2) reporting of the methods and results, 3) translating the authors findings (discussion section), and 4) writing style.

First, it is unclear how health services costs and patient costs were aligned in the overall cost assessment performed from the societal perspective. The authors do not clearly define and report what was their study population for which data on costs were ascertained. More specifically, it is hard to tell whether the patient cohort assessed for health systems costs were same (or different) from that of patient costs. For example, the authors report that participation in the study was voluntary, which this reviewer assumed that this was for the patient cost survey. If this assumption is correct, the authors are combining costs from the two different perspectives using two different cohorts/population, which is methodologically inappropriate.

Continuing with the methodological issue, the authors’ assessment of cost-effectiveness is not based on a randomized controlled study data and is of a relatively small sample of patients. The authors provide the distribution of socio-demographic and clinical characteristics between the two groups and show that they are not statistically significant as a way to justify appropriateness to ascertain true incremental costs and effectiveness between the two strategies. However, because this study was not done using an experimental design nor used appropriate methods to adjust for potential confounding factors, we cannot rule out the issue that the balance in these characteristics were observed by ‘chance’. Furthermore, there are a range of additional factors that can influence patient’s treatment outcomes such as severity of the disease at the time of treatment initiation (e.g., smear status, drug susceptibility profile, past TB history, co-morbidities other than HIV, CD4 count). Without exploring these factors and how these are balanced between the two groups, it is difficult claim that treatment success and cost-savings are actually achieved solely due to the implementation of home-based care.

Second, reporting of results in the tables and the text are very difficult to comprehend. For example, in Table 3, the authors report frequencies for each treatment component variables for the respective strategy, but numbers are reported as 600/594 for ‘duration of MDR-TB treatment (days) for those in the routine care. What does this mean? Does this mean the MDR-TB patients in the routine care were treated on average 600 days? If so, what does 594 mean? Similar issues are observed in other tables. In Table 4, service component costs and time estimates are reported together in a single table. Because the authors do not provide units of measurement, it can only be assumed that variables such as ‘time taken hospitalized’ is time associated costs and this was $101 for clinic-based strategy and $58 for home-based care. Is this correct? If so, authors need to clearly present these data with detailed text that shows how costs were calculated for each component. One suggestion on this front is to provide a separate result table for time-associated costs that were ascertained from patient cost survey (separately reported for time data and costs).

Third, the authors rarely discuss the findings of their study in the discussion section. This is an important section where the authors can 1) summarize their findings beyond reporting their quantitative results, 2) translate their findings so that readers can contextualize this research work to the issues in MDR-TB care and other interventions. For example, the authors in the third paragraph (starting from line 343) describe the cost trade-offs in patients’ travel costs between the two interventions, but do not provide actual cost estimates that were calculated from their study (need a statement like… “cumulative travel expenses incurred by patients in the home-based care were $XX lower than those reported by the patients in the clinic-based strategy.”). For paragraph starting in Line 371, what part of the authors findings ‘insinuate’ the implementation of interventions to alleviate financial barriers (and future research on the topics of social protection package)? This is a very bold claim to have without properly justifying/backing with the study data. Lastly (but not all), there are many limitations of this study (above mentioned issues are just a few major issues noted by this reviewer) that the authors need to carefully address in the discussion section. These limitations (methodology, types of data included, selection of the study population, etc.) have major implications on how readers would interpret the author’s study findings.

Lastly, this reviewer feels that the authors focus on the costing part of the study. The reason for this is that the methods and approach the authors have used does not meet the standards of cost-effectiveness analysis. Instead, the authors should consider a cost consequence analysis. In doing so, the authors should clearly present cost-trade offs between patients in the respective groups of the intervention (and also reflecting some of the issues raised above). The authors should also clearly describe results from the patient cost survey, separately reviewing detailed components of the survey outcomes and clearly describing numbers of participants reporting each question/data, uncertainty ranges, and median/mean estimates for each parameter, including time estimates. The authors can present the effectiveness data (improve their current Table 5), but not claim causality in costs and effectiveness resulting from home-based care as presented in Table 6. Authors sensitivity analyses can focus on describing factors that can contribute to the uncertainties in per-patient cost estimates for health systems and patient costs (separately). Ultimately, the authors need to vastly improve their discussion section as described above to better translate their findings to the relevant contexts where home-based care for MDR-TB can be implemented.

6. PLOS authors have the option to publish the peer review history of their article (what does this mean?). If published, this will include your full peer review and any attached files.

Reviewer #1: No

---

## [Author Response · Author response to Decision Letter 0]

19 Nov 2022

Centre for Development Support

Faculty of Economic and Management Sciences

University of the Free State

P.O. Box 399

Bloemfontein 9300

South Africa 

Academic Editor

PLOS ONE

04 November 2022

REF: Revision and resubmission of manuscript PONE-D-22-05265] - [EMID:5a0952a3e18fc9a6]

Dear Editor

Thank you for your email and the opportunity to revise our paper on ‘Cost- effectiveness of ambulatory clinic- and home-based multidrug-resistant tuberculosis management models in Eswatini.’ The suggestions offered by the reviewer have been immensely helpful in improving some important aspects of the manuscript. 

We have included the academic editor and reviewers’ comments (in italics) immediately after this letter and responded to them individually, indicating exactly how we addressed each comment and describing the changes we have made. The revisions have been approved by all authors. The changes in the revised manuscript accompanying this letter are marked up and all page numbers refer to the revised manuscript file with tracked changes.

We hope the revised manuscript will better suit PLOS ONE, but will be happy to consider further revisions and we thank you for your continued interest in our research.

Sincerely,

Ernest Peresu 

Academic Editor and Reviewer Comments and Author Responses 

Academic Editor

Comment 1: Please ensure that your manuscript meets PLOS ONE's style requirements, including those for file naming.

Response: Thank you for this reminder. We have revised our manuscript to ensure that it aligns with PLOS ONE's style requirements. 

Comment 2: We note that you have indicated that data from this study are available upon request. In your revised cover letter, please address the following prompts:

Response: Thank you for reminding us about the importance of making data from this study available. We have uploaded the minimal anonymised data set as a Supporting Information file (S2 File).

Comment 3: Your ethics statement should only appear in the Methods section of your manuscript. If your ethics statement is written in any section besides the Methods, please delete it from any other section.

Response: We highly appreciate your comment. The ethics statement now only appears in the Methods section of our manuscript. 

Comment 4: 4. Please include captions for your Supporting Information files at the end of your manuscript, and update any in-text citations to match accordingly. Please see our Supporting Information guidelines for more information: http://journals.plos.org/plosone/s/supporting-information.

Response: Thank you for this reminder and we have ensured that our in-text citations match captions of Supporting Information files. 

Reviewer 1

General comment: The research topic explored by the authors of this manuscript is an important area where there is not a lot of empiric evidence. The authors provide a good range of empiric evidence on costs using a wide range of data sources where costs can be ascertained at individual patient level from the societal perspective (partial). However, this reviewer has several major concerns for 1) the methodology in evaluating the ‘cost-effectiveness’, 2) reporting of the methods and results, 3) translating the authors findings (discussion section), and 4) writing style.

Response: We would like to thank the reviewer for carefully and thoroughly reading our manuscript and for the thoughtful comments and insightful suggestions. Drawing from the reviewer’s constructive comments, we have restricted the economic analysis in this study to a cost-consequence analysis instead of the previous cost-effectiveness evaluation. We are indeed grateful to the reviewer for his positive and encouraging comments.

Comment 1: First, it is unclear how health services costs and patient costs were aligned in the overall cost assessment performed from the societal perspective. The authors do not clearly define and report what was their study population for which data on costs were ascertained. More specifically, it is hard to tell whether the patient cohort assessed for health systems costs were same (or different) from that of patient costs. For example, the authors report that participation in the study was voluntary, which this reviewer assumed that this was for the patient cost survey. If this assumption is correct, the authors are combining costs from the two different perspectives using two different cohorts/population, which is methodologically inappropriate.

Response: Thank you for pointing this out. We found this comment very helpful. We have added more clarity on how data on costs from the health system and patient perspective was obtained from the 2014 and 2017 heterogeneous cohorts respectively (line 164 – 166) for results reported herein to be considered in light of this limitation. 

Comment 2: Continuing with the methodological issue, the authors’ assessment of cost-effectiveness is not based on a randomized controlled study data and is of a relatively small sample of patients. The authors provide the distribution of socio-demographic and clinical characteristics between the two groups and show that they are not statistically significant as a way to justify appropriateness to ascertain true incremental costs and effectiveness between the two strategies. However, because this study was not done using an experimental design nor used appropriate methods to adjust for potential confounding factors, we cannot rule out the issue that the balance in these characteristics were observed by ‘chance’. Furthermore, there are a range of additional factors that can influence patient’s treatment outcomes such as severity of the disease at the time of treatment initiation (e.g., smear status, drug susceptibility profile, past TB history, co-morbidities other than HIV, CD4 count). Without exploring these factors and how these are balanced between the two groups, it is difficult claim that treatment success and cost-savings are actually achieved solely due to the implementation of home-based care. 

Response: This is a great observation and indeed true. The findings of this study are limited by the inherent limitations of the study design, in particular, the small sample size. We concur that even though the two groups have relatively similar socio-economic conditions and negligible dissimilarity in clinical characteristics, there are confounding factors that can still influence the analyses and interpretation of our findings. However, we consider that in light of the above, any effect on our results is likely to be modest in this study. Nonetheless, to allow readers to interpret our results with caution, we have now emphasised this potential limitation in the revised discussion in line 431 – 439 as follows:

“Patients in the clinic-based model had generally longer periods of hospitalisation (41 days), particularly during the intensive phase, than those under home-based care (23 days). Consequently, inpatient care was the main contributor to health system costs (32%) in the clinic-based model. Our analysis did not consider the severity of the disease at the time of treatment initiation or delays in initiating MDR-TB treatment after diagnosis. This relatively higher rate of hospitalisation may be taken as a proxy measure of severe disease presentation by patients in the Lubombo region. The higher risk of unfavourable outcomes among those patients delayed initiating MDR-TB treatment has been consistently reported in studies from elsewhere.22”

Comment 3: Second, reporting of results in the tables and the text are very difficult to comprehend. For example, in Table 3, the authors report frequencies for each treatment component variables for the respective strategy, but numbers are reported as 600/594 for ‘duration of MDR-TB treatment (days) for those in the routine care. What does this mean? Does this mean the MDR-TB patients in the routine care were treated on average 600 days? If so, what does 594 mean? Similar issues are observed in other tables. In Table 4, service component costs and time estimates are reported together in a single table. Because the authors do not provide units of measurement, it can only be assumed that variables such as ‘time taken hospitalized’ is time associated costs and this was $101 for clinic-based strategy and $58 for home-based care. Is this correct? If so, authors need to clearly present these data with detailed text that shows how costs were calculated for each component. One suggestion on this front is to provide a separate result table for time-associated costs that were ascertained from patient cost survey (separately reported for time data and costs). 

Response: We appreciate the reviewer’s insightful suggestion and we have modified and improved Table 3 by presenting frequencies (mean and median) for each treatment component variables for the respective strategy separately. Our original Table 4 reported service component costs and time estimates together in a single table. To facilitate easy interpretation of direct costs and time-associated costs components for patients and caregivers, we have incorporated the reviewer’s suggestion by presenting these separately in Table 5 and 6 respectively, with direct and opportunity cost of time components clearly disaggregated. In addition, we also added a detailed account of how these costs were calculated for each component in Line 215 – 231. 

Comment 4: Third, the authors rarely discuss the findings of their study in the discussion section. This is an important section where the authors can 1) summarize their findings beyond reporting their quantitative results, 2) translate their findings so that readers can contextualize this research work to the issues in MDR-TB care and other interventions. For example, the authors in the third paragraph (starting from line 343) describe the cost trade-offs in patients’ travel costs between the two interventions, but do not provide actual cost estimates that were calculated from their study (need a statement like… “cumulative travel expenses incurred by patients in the home-based care were $XX lower than those reported by the patients in the clinic-based strategy.”). For paragraph starting in Line 371, what part of the authors findings ‘insinuate’ the implementation of interventions to alleviate financial barriers (and future research on the topics of social protection package)? This is a very bold claim to have without properly justifying/backing with the study data. Lastly (but not all), there are many limitations of this study (above mentioned issues are just a few major issues noted by this reviewer) that the authors need to carefully address in the discussion section. These limitations (methodology, types of data included, selection of the study population, etc.) have major implications on how readers would interpret the author’s study findings. 

Response: Thank you for reminding us how important it is to discuss the actual findings that were calculated from our study in the discussion section. We found your comments extremely helpful and have revised accordingly and included several actual cost estimates from our study to aid our discussion. We have also rephrased paragraph starting in Line 371 (original manuscript) and it now reads:

“However, it remains unclear to what extent, if any, the superior treatment success observed in the home-based strategy can be attributed to the availability of the social protection packages. Further research can evaluate the influence of these social protection packages on patient healthcare expenditures and treatment outcomes.”

You also make a valid point that the paper should focus more explicitly on meaningfully presenting the study limitations. We have now provided a more complete presentation of the study’s limitations (line 501 – 513) and their potential limitation and where applicable, steps taken to mitigate them. 

Comment 5: Lastly, this reviewer feels that the authors focus on the costing part of the study. The reason for this is that the methods and approach the authors have used does not meet the standards of cost-effectiveness analysis. Instead, the authors should consider a cost consequence analysis. In doing so, the authors should clearly present cost-trade offs between patients in the respective groups of the intervention (and also reflecting some of the issues raised above). The authors should also clearly describe results from the patient cost survey, separately reviewing detailed components of the survey outcomes and clearly describing numbers of participants reporting each question/data, uncertainty ranges, and median/mean estimates for each parameter, including time estimates. The authors can present the effectiveness data (improve their current Table 5), but not claim causality in costs and effectiveness resulting from home-based care as presented in Table 6. Authors sensitivity analyses can focus on describing factors that can contribute to the uncertainties in per-patient cost estimates for health systems and patient costs (separately). Ultimately, the authors need to vastly improve their discussion section as described above to better translate their findings to the relevant contexts where home-based care for MDR-TB can be implemented. 

Response: Thank you for your thorough review and salient observations. We agree with the reviewer’s assessment. It is our sincere hope that this study provides the necessary evidence to guide decision makers in assessing the resource use and costs and health outcomes for the two ambulatory models of MDR-TB care and possibly generate hypotheses for definitive cost-effectiveness studies. 

To further balance the implications of our results with the potential limitations of the study design, we have adopted a cost-consequence analysis approach to our study as per your suggestion. Accordingly, we have changed the title to “Cost-consequence analysis of ambulatory clinic- and home-based multidrug-resistant tuberculosis management models in Eswatini.” Furthermore, throughout the manuscript, we have replaced the term cost-effectiveness with cost-consequence. 

We have accordingly made revisions, clearly describing findings from the patient cost survey, presenting disaggregated direct and opportunity time cost components, improving the reporting of frequencies for health system components in tables, illustrating treatment outcome results (line 352 – 356; 366 – 370) and performing a sensitivity analysis directed on some of the uncertainties in per-patient cost estimates for health systems and patient costs (line 383 – 394). The discussion section has been improved by highlighting key findings on costs, placing the results in context and, where necessary, noting how they may affect the validity of our study (lines 407 – 409; 422 – 423; 431 – 434; 444 – 451; 466 – 468; 481 – 483; 501 – 513).

---

## [Decision Letter · Decision Letter 1]

27 Jul 2023

PONE-D-22-05265R1Cost-consequence analysis of ambulatory clinic- and home-based multidrug-resistant tuberculosis management models in EswatiniPLOS ONE

Dear Dr. Peresu,

Thank you for submitting your manuscript to PLOS ONE. After careful consideration, we feel that it has merit but does not fully meet PLOS ONE’s publication criteria as it currently stands. Therefore, we invite you to submit a revised version of the manuscript that addresses the points raised during the review process.

Please see some additional minor comments from the previous reviewer below. One new reviewer assessed the manuscript, and found no concerns that require addressing.

We look forward to receiving your revised manuscript.

Kind regards,

Hanna Landenmark

Staff Editor

PLOS ONE

Journal Requirements:

2) Please include a complete copy of PLOS’ questionnaire on inclusivity in global research in your revised manuscript. Our policy for research in this area aims to improve transparency in the reporting of research performed outside of researchers’ own country or community. The policy applies to researchers who have travelled to a different country to conduct research, research with Indigenous populations or their lands, and research on cultural artefacts. The questionnaire can also be requested at the journal’s discretion for any other submissions, even if these conditions are not met.  Please find more information on the policy and a link to download a blank copy of the questionnaire here: https://journals.plos.org/plosone/s/best-practices-in-research-reporting. Please upload a completed version of your questionnaire as Supporting Information when you resubmit your manuscript.

Reviewers' comments:

Reviewer's Responses to Questions

**Comments to the Author**

1. If the authors have adequately addressed your comments raised in a previous round of review and you feel that this manuscript is now acceptable for publication, you may indicate that here to bypass the “Comments to the Author” section, enter your conflict of interest statement in the “Confidential to Editor” section, and submit your "Accept" recommendation.

Reviewer #1: (No Response)

Reviewer #2: All comments have been addressed

2. Is the manuscript technically sound, and do the data support the conclusions?

Reviewer #1: Yes

Reviewer #2: Yes

3. Has the statistical analysis been performed appropriately and rigorously? 

Reviewer #1: Yes

Reviewer #2: Yes

4. Have the authors made all data underlying the findings in their manuscript fully available?

Reviewer #1: Yes

Reviewer #2: Yes

5. Is the manuscript presented in an intelligible fashion and written in standard English?

Reviewer #1: Yes

Reviewer #2: Yes

6. Review Comments to the Author

Reviewer #1: Thank you for addressing my comments in your revision - a lot of hard work went into this work.

I have following list of minor revision request for the authors upon review of the revised version:

1) For the sentence starting “The higher risk of unfavourable outcomes among those patients delayed initiating MDR-TB treatment has been consistently reported in studies from elsewhere.22”

Please indicate two to three examples of unfavourable outcomes associated with delayed treatment initiation of MDR-TB.

2) Table 3: rename the table title as “summary of key treatment management indicators for MDR-TB patients” and report (for each component and for each intervention) sample size used to calculate each summary statistics. Also, summary statistics can either be median or mean (based on the distribution of the data) and the authors should report uncertainty ranges (95% CI or IQR).

3) Line 515-521: How does no including cost for laboratory investigation in this study relates to Pooran et al.,’s finding? Since this is a cost consequence analysis, not including certain key health care cost estimates essentially under-estimates actual cost (for each arm) of MDR-TB care. In cost-effectiveness analysis, these costs may cancel out in calculating ICER, but in this analysis, providing an overall cost estimate is important as the authors are not calculating ICER. Authors should state that the limitations in not including key health care costs for MDR-TB care, especially on diagnostics, under-estimates the overall MDR-TB care costs. If authors can provide specific items that are not included, they can suggest by how much (approximately) the authors MDR-TB care estimates is under-estimated.

4) Line 526-528: Strength comment as a separate paragraph may not be necessary as it’s being discussed in the earlier section. Please delete.

5) Line 532: delete “although qualified by the study limitations” (not necessary).

Reviewer #2: The manuscript is well written and understandable to me. I hope the manuscript will add a valuable contribution in the literature.

7. PLOS authors have the option to publish the peer review history of their article (what does this mean?). If published, this will include your full peer review and any attached files.

Reviewer #1: No

Reviewer #2: **Yes: **Fahmida Rahman

---

## [Author Response · Author response to Decision Letter 1]

3 Feb 2024

Reviewer Comments and Author Responses 

Reviewer 1

General comment: Thank you for addressing my comments in your revision - a lot of hard work went into this work.

Response: We would like to thank the reviewer for carefully and thoroughly reading our manuscript and for the thoughtful and encouraging comments. We have incorporated these insightful suggestions to improve our paper. 

Comment 1: For the sentence starting “The higher risk of unfavourable outcomes among those patients delayed initiating MDR-TB treatment has been consistently reported in studies from elsewhere.” Please indicate two to three examples of unfavourable outcomes associated with delayed treatment initiation of MDR-TB.

Response: Thank you for pointing this out. We found this comment very helpful. We have added two unfavourable outcomes associated with delayed initiation of MDR-TB treatment (line 401 – 404). The sentence now reads:

“The elevated risk of unfavourable outcomes such as high mortality and increased person-to-person transmissions among those patients who delayed initiating MDR-TB treatment has been consistently reported in studies from elsewhere.” 

Comment 2: Table 3: rename the table title as “summary of key treatment management indicators for MDR-TB patients” and report (for each component and for each intervention) sample size used to calculate each summary statistics. Also, summary statistics can either be median or mean (based on the distribution of the data) and the authors should report uncertainty ranges (95% CI or IQR).

Response: We appreciate the reviewer’s insightful suggestion and we have renamed Table 3 (line 282 – 283). We have also modified and improved Table 3 by presenting the mean and the associated 95% confidence interval for each treatment component variable for the respective strategy.

Comment 3: Line 515-521: How does no including cost for laboratory investigation in this study relates to Pooran et al.,’s finding? Since this is a cost consequence analysis, not including certain key health care cost estimates essentially under-estimates actual cost (for each arm) of MDR-TB care. In cost-effectiveness analysis, these costs may cancel out in calculating ICER, but in this analysis, providing an overall cost estimate is important as the authors are not calculating ICER. Authors should state that the limitations in not including key health care costs for MDR-TB care, especially on diagnostics, under-estimates the overall MDR-TB care costs. If authors can provide specific items that are not included, they can suggest by how much (approximately) the authors MDR-TB care estimates is under-estimated. 

Response: This is an excellent comment, and we agree that diagnostic costs are any important component that should be included in a cost-consequence analysis assessment study. In this study, we indeed considered and reported the cost of laboratory diagnostic and monitoring tests in supplementary S1 Table (line 190); in text (line 292 – 296) and in Table 4 (line 298 – 300). The total laboratory test costs were determined by multiplying unit costs by the frequency of tests performed for the period of MDR-TB treatment as recommended in the national TB policy guidelines. Given that the unit cost of each component associated with diagnosis of MDR-TB in Eswatini was not readily available from the Eswatini National TB Control Programme (NTCP), our study used unit cost data reported by Pooran and colleagues in a study conducted in South Africa titled “What is the Cost of Diagnosis and Management of Drug Resistant Tuberculosis in South Africa?”. There is negligible variation in MDR-TB clinical practices, life expectancy, payment systems and discount rate between Eswatini and South Africa – in a way enabling the geographic transferability of unit cost data for each laboratory diagnostic component. 

Comment 4: Line 526-528: Strength comment as a separate paragraph may not be necessary as it’s being discussed in the earlier section. Please delete. Line 484-486. 

Response: This is a great observation and indeed true. We have deleted the strength comment (line 484 – 486).

Comment 5: Line 532: delete “although qualified by the study limitations” (not necessary). Line 489. 

Response: Thank you for this suggestion. We have deleted “although qualified by the study limitations” (line 489).

---

## [Editor Report · Decision Letter 2]

19 Mar 2024

Cost-consequence analysis of ambulatory clinic- and home-based multidrug-resistant tuberculosis management models in Eswatini

PONE-D-22-05265R2

Dear Dr. Peresu,

We’re pleased to inform you that your manuscript has been judged scientifically suitable for publication and will be formally accepted for publication once it meets all outstanding technical requirements.

Kind regards,

Pracheth Raghuveer, MD, DNB

Academic Editor

PLOS ONE

---

## [Editor Report · Acceptance letter]

22 Mar 2024

PONE-D-22-05265R2 

PLOS ONE

Dear Dr. Peresu, 

I'm pleased to inform you that your manuscript has been deemed suitable for publication in PLOS ONE. Congratulations! Your manuscript is now being handed over to our production team.

Kind regards, 

on behalf of

Dr. Pracheth Raghuveer 

Academic Editor

PLOS ONE